# Improving the Management of Children with Fevers by Healers in Native Rural Areas in the South of Ecuador

**DOI:** 10.3390/ijerph20053923

**Published:** 2023-02-22

**Authors:** Estefanía Bautista-Valarezo, Maria Elena Espinosa, Narciza Eugenia Arce Guerrero, Veronique Verhoeven, Kristin Hendrickx, Nele R. M. Michels

**Affiliations:** 1Facultad de Ciencias de la Salud, Universidad Técnica Particular de Loja, Loja 1101608, Ecuador; 2Department of Family Medicine and Population Health, School of Medicine and Health Sciences, University of Antwerp, Universiteitsplein 1, 2610 Antwerp, Belgium; 3Programa de Doctorado en Ciencias Médicas, Universidad de la Frontera, Temuco 4811230, Chile; 4Facultad de Ciencias Médicas, Universidad de Cuenca, Cuenca 010203, Ecuador

**Keywords:** children, fever, alarm signs, socioeconomic factors, traditional medicine

## Abstract

Background: Indigenous populations are represented among the poor and disadvantaged in rural areas. High rates of infectious diseases are observed in indigenous child populations, and fever as a general symptom is common. Objective: We aim to improve the skills of healers in rural indigenous areas in the South of Ecuador for managing children with fevers. Method: We performed participatory action research (PAR) for this study with 65 healers. Results: The PAR focused on the following four phases: (1) ‘observation,’ eight focus groups were used. (2) ‘planning’ phase was developed, and with culturally reflective peer group sessions, a culturally adapted flowchart was constructed titled “Management of children with fever.” In phase (3): ‘action’, the healers were trained to manage children with fever. Phase (4): ‘evaluation’, 50% of the healers used the flowchart. Conclusions: Explicit recognition of the need for traditional healers and health professionals in indigenous communities to work together to improve health indicators such as infant mortality exists. Additionally, strengthening the transfer system in rural areas is based on knowledge and cooperation between the community and the biomedical system.

## 1. Introduction

One of the Millennium Development Goals of the United Nations (UN) was to reduce mortality in children below 5 years old by two-thirds between 1990 and 2015. The UN 2015 report signals that the mortality rate has reduced by half, from 90 to 43 deaths per 1000 live-born children between 1990 and 2015 [1]. Despite the reduction in child mortality, there is still room for improvement, especially in low- and middle-income countries where a number of socioeconomic determinants such as poverty, schooling/illiteracy, and ethnic groups still cause millions of deaths in children below 5 years old [2,3].

These determinants are seen as barriers that hinder access to basic clinical facilities, especially in rural areas. An ecological study on child mortality associated with socioeconomic factors in Ecuador reviewed three population censuses conducted by the National Institute of Statistics and Censuses (Instituto Nacional de Estadística y Censos, INEC) (1990, 2000, 2010) and mentioned a 5.1% increase in child mortality in the proportion of the indigenous ethnic group when comparing the years 2010 and 2000 [4].

In a study on indigenous health, Gracey and King (2009) mention that certain determinants such as deficient life conditions, inadequate nutrition, and exposure to high infection rates are considered enough reasons for the presence of health inequities in indigenous populations [5]. These inequities are mainly reflected in the worsening of health in women and children, presenting high rates of mortality, child mortality, and infectious diseases [6].

In rural indigenous communities, traditional healers are a key component in providing care, thus becoming a complement to the biomedical-oriented health system. The healers’ knowledge is influenced by their roots, tradition, and culture, which allows them better access to this indigenous population. They also are knowledgeable in the properties of plants and in the diagnosis of some diseases, especially from their culture [7,8]. A clear example is the study carried out in the Kilosa and Handen districts of Tanzania which demonstrated the critical role of traditional healers in providing “biomedically accepted first aid,” leading to the lowering of body temperature and thus increased chances of survival of patients with malaria (children) [9]

However, some studies raise important questions about how traditional healers manage high-risk diseases. In a retrospective study in Malawi, the researchers note that care from traditional healers was associated with increased mortality after burns in children. Likewise, other studies mention that in places with pluralistic medical care, where informal health resources such as traditional healers or pharmacies can be used, the prior use of these services can delay medical diagnosis and treatment as many symptoms of diseases are not perceived as severe or attributed to traditional causes [10,11,12].

In several countries such as Ecuador, traditional medicine is an important part of the health system. This fact leads to the need to create bridges between the biomedical-oriented health staff and the traditional-oriented healers in order to improve community health care. One of the key aspects is creating collaboration, reliable communication, and appropriate referral systems [13]. The training of traditional healers generates positive changes in the care they provide to their patients, as Hoff (1992) mentions in his study on community health. Additionally, he signals a better predisposition of traditional healers to working with biomedical staff [8,14].

This study focused on the need to explore the management of children with fever in southern Ecuador and to focus development skills on recognizing alarm signals in children. In collaboration with medical professionals, a referral plan is developed and supported by a culturally adapted flowchart. The main items were the recognition of alarm signs in children with fever and the creation of bridges that can contribute to the collaboration between the traditional health system and the biomedical one. This paper is part of a broader project that studies the aspects of intercultural health care in the communities of southern Ecuador [15]. This project arises from the need for a health system whose priority is to recognize the barriers that hinder teamwork between traditional healers and health staff. This knowledge creates tools that allow better collaboration and coverage in care in rural indigenous areas in Ecuador. In an earlier stage, Ecuadorian indigenous healers’ perceptions about health and illness and the barriers that compromise the relationship between traditional healers and health staff were investigated [12,16].

## 2. Methods

To reach our goal, participatory action research (PAR) was used. The PAR focuses on a reflection and joint action process to implement changes and generate theory [17]. It also seeks critical self-reflection and the ability to think by the participants. There are several kinds of PAR; our approach is related to the “empowerment type” [18] because it explores and involves members of the community taking part in the research activity [19]. The study comprises four phases: (1) observation, (2) planning, (3) acting, and (4) assessment, which will be explained further on in the manuscript [20]. Our study was conducted over a period of four years, from September 2016 to December 2021.

In this study, through the PAR methodology, indigenous knowledge initially through collective inquiry allowed knowing the perceptions and management of children with fever in indigenous communities [21]. Subsequently, through a reflective process between the researchers and the participants, a tool was created that is the culturally adapted flowchart. This tool is focused on the early recognition of warning signs in children with fever and the management of a referral plan [22].

This paper follows the consolidated criteria for reporting qualitative research (COREQ) guidelines for reporting qualitative research.

### 2.1. Ethical Approval

The ethical requirements of the Research Committee of the University of San Francisco de Quito (CEISH USFQ 2017-059E) were followed. The written consent of the indigenous community (the “Saraguro and Cuenca” Healers’ Council) was obtained, as well as that of the participants, who were free to refuse to participate or to withdraw at any given time.

### 2.2. Phase 1: Observation

In the observation phase, we intended to understand the organizational culture under which traditional healers work and how it is related to the biomedical health system. The data analysis was based on a phenomenological method: this means that we focused on participants exploring, describing, and understanding their experiences with a phenomenon: in this case, caring for a child with fever in indigenous populations [23].

#### 2.2.1. Purposive Sampling Was Used to Recruit the Participants

In order to ease discussions at various levels, different types of healers participated in the research [24]. The traditional healers selected were midwives, chamanes (“Yachay”/”Uwishin” or “visionarios”), and herb healers (“Hierbateros”), from the Saraguro and Cuenca regions. In the first phase of the research, the approach to the indigenous community was carried out specifically to the traditional healers through two primary healthcare technicians from the same community. We chose to recruit indigenous groups from southern Ecuador because: (a) these communities are the center of meeting for indigenous celebrations, and (b) these communities belong to native Ecuadorian people. All the participants were renowned healers in their communities and expressed experience and interest in the study. Additionally, the participants needed to be able to communicate in Spanish, although, for the healers, it was not their mother language, which explains the grammatical mistakes that you might find in the quotes.

#### 2.2.2. Procedure

In the focus groups, clinical cases on paper were presented to and discussed by the participants, to help understand the management of children with fever from the perspective and context of traditional healers. The clinical cases were written clearly and concisely, and the simplicity of the language allowed the participants to have better interaction and understanding. The main idea focused on the warning signs in children with fever. Each focus group was conducted with the help of one moderator (researcher) and one observer (researcher). The guide for the focus groups was elaborated by the team of researchers and the healers. This consisted of a clinical case focusing on the management of children with fever in rural indigenous communities (Table 1). In addition, their informed consent and authorization to audio-record the discussions were asked for.

Content analysis with a phenomenological approach was used to analyze the data and identify themes, as well as to come to a final agreement on the results. The data recorded from the focus group and the observation notes were transcribed verbatim and analyzed with the N-Vivo software version 11. Three researchers explored and evaluated the data to structure and describe the patterns which were categorized and coded. Two of the researchers speak Spanish and come from the southern region of Ecuador, so they know the worldview of the place.

The code book was shared among the researchers, who later correlated the results to learn more about the problems of the management of children with fever [25,26]. The phenomenological analysis focused on looking at the experiences lived by the participants, reflecting on the action in each of the clinical cases, and interpreting through a clear and interpretative discourse [27]. In the results, the perception of the healers was presented because it seeks to know the ideas, skills, and aptitudes of the healers about the management of children with fever, and with this content, design a culturally adapted tool (flow chart) that allows them to strengthen their work and create communication bridges with the biomedical system to improve health in indigenous communities.

### 2.3. Phase 2: Planning (Creation of a Plan)

In this second phase, we sought to create knowledge based on the observation results in order to solve the known problems. The qualitative results (of phase 1) were sent and presented again to all the involved researchers for a critical analysis. In three reflexive group sessions, a flowchart of children with fever was developed. As the results took into account the opinion of healers, we can say it is a culturally adapted flowchart. Reflexibility was an important cornerstone in the validity and transparency of the research and allowed the researchers to have constant interaction with the phenomenon under study [28]. The goal was to establish a joint strategy to improve the practices and the inter-cultural communication bridges. Additionally, the strategy focused on the detection of alarm signs in children with fever and activating a referral system in a rural health community team. The researchers created the flowcharts based on the results obtained in phase 1, and the design was created with the help of a graphic designer. The design of the graphics was based on the Saraguro indigenous culture, which is the participants’ place of birth and residence. The designer carried out a prior revision of the indigenous group. Finally, a last reflexive group session was held with the traditional healers; the flowchart was presented and discussed. The healers made some suggestions on the content and the graphics of the flowchart presented such as warning signs and added words such as purple or bluish skin to facilitate the interpretation of cyanosis.

### 2.4. Phase 3: Action Plan

In the third phase, we sought to train the traditional healers in the care of children with fever and to implement the culturally adapted flowchart. We also sought to develop skills that allow the healers to recognize the alarm signs and to work with the health professionals in a referral system when the patient requires so. This phase is dynamic and focused on the results from the previous phases. The health professionals involved in the training were the researchers who are one family doctor, one pediatrician, and one obstetrician–gynecologist. It is clear that diverse processes are linked, such as: (1) the healers’ perceptions and beliefs, (2) the critical and participatory analysis among researchers and healers, and (3) the sharing of knowledge, skills, and resources to improve teamwork in the community.

Workshops were organized with healers and health professionals, working together on the management of children with fever, hereby using the culturally adapted flowchart. The training had three important points: (1) the use of a thermometer and its interpretation, (2) the recognition of alarm signs and patient management (steps from the flowchart), and (3) the patient referral system. Each workshop was recorded with the prior consent of each participant.

### 2.5. Phase 4: Evaluation

One year after the training of the traditional healers and health professionals, the researchers approached the indigenous communities again to assess the implementation of the culturally adapted flowchart in children with fever through a survey.

Each of the questions asked sought to assess the following: (1) the level of skill learning and the performance of the healers in recognizing the warning signs in children with fever, (2) the management (diagnoses and treatment) of children with fever, and (3) the referral of children with fever to the health center or to the hospital (Table 2). The assessment instrument was a survey that allowed deepening on the aforementioned points. The descriptive analysis of the results was carried out through central tendency and standard deviation for quantitative variables and frequency and percentage for qualitative variables.

## 3. Results

### 3.1. Phase 1: Observation

Eight focus groups were assembled, which were conducted with 65 participants aged between 28 and 70 years old. The healers belong to the Kichwa, Shuar, and *Cholos Cuencanos* ethnic groups, working as shamans (“yachay”/“uwishin” or “Visionaries”), herb-healers (“weedlers”), midwives (“midwives”), or bonesetters (“sobadores”). (Table 3)

Focus groups lasted between 80 and 140 min, plus 30 min for initial orientation and engagement with the participants. Code saturation was achieved in the third focus group; the remaining focus groups were analyzed until meaning saturation was attained in all the codes. The saturation criteria of the discourse were the repetition of the topics and that no new information was shared. Despite the criteria, saturation is an art that will depend on the type of questions and how they are asked to obtain different answers. In addition, the nuance of an answer can have different interpretations; here, the importance of triangulation arises.

#### 3.1.1. Main Themes

The researchers presented the participants with three clinical cases, with questions based on the search for warning signs and the management of children with fever. Participants mentioned their positive experiences, joint problems, and solutions regarding clinical case management. The themes that emerged after presenting the clinical cases could be classified into four main categories (Table 4).

##### Caring for Children with Fever in Indigenous Communities

Traditional healers provide most of their care services in their own homes, or they go to the patient’s home. In the beginning, care focuses on a series of questions targeted to the child’s caregiver and which are focused on the time that the child has had a fever, accompanying signs/symptoms, and medications administered (Q1). They also ask questions aimed at seeking pathologies inherent to the indigenous culture, for example, “mal aire” (“bad air”), “el espanto” (“the fright”), or “el mal de ojo” (“the evil eye”) (Q2). The attention of the doctors in the indigenous communities is given in the rural health centers through external consultation; they usually make home visits.


*“Well, I believe that first (……) the research is done with the mother of the family, from the moment the child has a fever; it’s already 5 days he’s got a fever, what medication or to where she has gone to cure the child, already (….), from the moment the child doesn’t go out or is not active inside the house”*
(Q1-FG4)


*“Suddenly the kid got wet being hot or suddenly he was asleep and woke up and like we say, the air caught him. The air catches him? He gets the air what we call he gets the air or the bad air, the cold; that the cold or the fright……..I clean him and give him some waters”*
(Q2-FG4)

In order to verify if the temperature is high or if the child has a fever, most of the healers use their hands; very few mention using a thermometer. They also add to this practice a detailed observation of the patient in search of signs which can warn of severity (Q3).


*“I heal with my hands, my hands heal….. I touch the child and see if he’s hot…..I’ve seen they use those thermometers….I cannot see there, I touch the child I feel the head, the belly, the little back.”*
(Q3-FG2)

Traditional healers mention that the treatment to reduce the temperature is to use physical means such as cloths with warm water and herbal water that the patient has to drink during the day. They also emphasize the importance of using certain types of herbs that they called “frescas” (“fresh”), as they allow for reducing the temperature and treating infections (Q4).


*“First we apply traditional medicine or ancestral medicine, we give fresh waters; for example mauve, begonia or chamomile waters, we perform a “bajeado” as we call it; the water boils, we place the flowers there, we put the lid on, the fragrance concentrates and you give it with a little bicarbonate.”*
(Q4-F2)

##### Alarm Signs in Children with Fever

During their care for a child with fever, traditional healers, both in their questions and in their exploration of the patient, are focused on looking for alarm signs or signs/symptoms that can help them find a diagnosis. The healers look for alarm signs in two steps: first, the questions are targeted at searching for a loss of appetite, physical inactivity due to feeling run-down, and an alteration of the state of consciousness. The second step consists of examining the patient in search of dehydration signs such as dry mucous membranes, dry skin, or alterations in the color of the skin, such as paleness or cyanosis (Q 5–6).


*“What I do is to see if the lips and the tongue are dry because they tell me he is dehydrated, but I also see if the child is run-down sad…”*
(Q5-FG3)


*“You watch the eyes, the tongue; the eyes fall down like small drops inside already, they become pale too, (…..) well let’s say the face is pale like hollow-cheeked and the mouth is totally dry and the tongue too; when this happens it means that the child has an infection or something that is really severe, because if the child is still fine he’s got energy.”*
(Q6-F5)

Other signs and symptoms that the healers ask about and explore are coughs, diarrhea, and headaches. When these signs are present together with fever, they suspect infectious processes.

##### Patient Follow-Up

Patient follow-up is one of the main characteristics of the care provided by traditional healers. On the first visit, they stay with the patient and provide their treatment to reduce the fever with physical means, and they prepare infusions with medical herbs. They also perform healing rituals focused on treating pathologies inherent to the culture; the most used rituals are “egg cleansing”, “diagnosis through urine”, and “blowing with medical herbs” (Q7–8).

Subsequent visits will be conducted according to the severity of the case, and the objective is to assess the fever, alarm signs, complications, and onset of new signs or symptoms. The healing process, as the healers call it, consists of attaining a balance between the physical, emotional, spiritual, and mental components of the patient, and it involves both the patient and the family environment (Q9).


*“You buy a little creole chicken egg, better if from the same day just laid…we make crosses with the egg….It’s like I say, it’s a ritual first asking the power from God, who is the one who has given us life first and the earth which is the part of the little animals that we use their live part, because the egg is a live part and that adsorbs gives its life so that that person really takes that life and alleviate.”*
(Q7-FG7)


*“I believe that he’s not going to come very soon so I have to follow him up closely and I have to go to the house all the same as they said to see how the house is if it’s clean how they eat sometimes.”*
(Q8-FG2)


*“For the child to heal the parents must be fine, ……there doesn’t have to be problems with the neighbor…..it’s a balance of the four bodies physical, emotional, spiritual, and mental of the patient, and it involves both the patient and the family environment.”*
(Q9-FG3)

##### Referral to the Health System

The healers assess the patient’s severity according to the duration of the fever, loss of appetite, alteration in mood, and dehydration of the child. Despite noticing the severity of a case, there are obstacles that prevent them from working in a team and referring the patient to the biomedical health system. The healers mention that these obstacles are the following: (1) the absence of logistics or coordination between the (biomedical) health system and the healers; (2) the distance between the indigenous communities and the hospitals; and (3) impaired communication bridges between the traditional healers and the health professional due to a clear perception of the physicians’ power and the criticism towards the community healers (Q10).

The complications they mention as reasons for transferring a patient to the health system are the following: severe dehydration, feverish convulsions, and alterations in the state of consciousness (Q11).


*“I once went to the hospital with a child and they didn’t let me in, they turned me out, that I know nothing, that I should see how the child is and I told them it is a severe infection.”*
(Q10-FG4)


*“Children with high fever start with convulsions then you do have to send them to a health post for them to help because it’s a very dangerous case if the child’s fever doesn’t go down quickly you already have to send him there.”*
(Q11-FG7)

### 3.2. Phase 2: Planning (Creation of a Plan)

#### Design of Flowchart in Management of Children with Fever

During approximately five months after phase one, discussion and interaction on the results with traditional healers found place. This was organized in small group meetings for the “dialogue of knowledge” or in their homes during care for children. The main points of focus to improve the management of children with fever were (1) warning signs, (2) treatment, and (3) referral to the health system. Although there was consensus on many aspects of the subject, different perspectives were also found. Several young healers mention the importance of knowing the warning signs in children with fever and identifying them. In contrast, the healers with more experience emphasize that by knowing the management of a child with fever, they have made it their priority and the greatest difficulty in handing the patient over to the health system. They mention the need to create bridges that allow them to work together and complement the biomedical health system with traditional health. Based on these results, group sessions among the researchers were held, and under several evaluation criteria, an instrument (flowchart) was designed that will allow traditional healers to recognize the alarm signs and the right moment to transfer the patient to the biomedical health system. Once the flowchart was finished, it was presented to the healers in a workshop, who suggested some changes that focused mainly on the design of the graphics and on the transfers to the health system (Figure 1).

### 3.3. Phase 3: Action Plan

A series of ten workshops were held on the management of children with fever flowchart facilitated by the researchers. These workshops were developed in culturally adapted skills laboratories; each group consisted of six people and lasted 2 h, and all healers were trained (*n* = 56). The main topics developed in the workshops were:The use of a thermometer and its interpretation;Recognition of alarm signs and patient management (steps from the flowchart);Patient referral system.

The content of the workshops responds to the needs of the healers observed during the investigation of children with fever.

Each healer received a folder containing the culturally adapted flowchart in the management of children with fever and a digital thermometer.

As the workshop progressed, the participants took on more active roles; the more experienced healers taught the younger ones about fever treatment with medicinal plants and the warning signs in children with fever. During the training, a greater acceptance could be observed about transferring the patient to the biomedical health system when warning signs appear.

The participants recognized the opportunity to acquire and share new knowledge. They assumed the responsibility of taking an active part in the communities for joint work with the biomedical health system. They also recognized the potential of learning through flow charts and how knowledge can be systematized in easily accessible primers.

### 3.4. Phase 4: Evaluation

A total of 60 healers were evaluated one year after the training. All the healers have seen at least 10 patients with fever in 12 months. Of them, 50% mention that they have used the flowchart at least on one occasion to recognize alarm signs or to transfer a patient. The alarm signs most observed by the healers have been the following: (1) signs of inactivity and (2) signs of dehydration. Additionally, only 15% of the healers mentioned having transferred their patients to the biomedical health system. Most of the transfers (70%) were accompanied by the healer. Finally, 60% of the healers think that the flowchart is moderately understandable and useful, and 5% think that it is poorly understood and not very useful.

## 4. Discussion

The generation of knowledge, learning, and adaptation on the management of children with fever in indigenous communities is one more step to building a more participatory health system between traditional medicine and the biomedical system. The strategy implemented was the implementation of culturally adapted flowcharts for healers from the south of Ecuador. The intervention of a diverse team (healers/health professionals/researchers) in research (PAR) addressed complex problems such as recognizing warning signs and transfers to the biomedical health system.

In a similar study by Kwame and colleagues (2021) on the integration between traditional medicine and the biomedical system, several approaches have been proposed, such as joint patient referrals, a collaboration between biomedical doctors and traditional healers, the creation of an intercultural health unit where the patient can choose the type of care, training given to healers, and the implementation of knowledge into the training of doctors. These integrative models are alternatives to improve care in rural areas and to reduce complications that are greater in populations with little access to health services [29].

In the indigenous communities, health differs from the rest of the population as concerns aspects such as culture, beliefs, ethnic group, language, nutrition, socioeconomic situation, and schooling level; these conditions should be taken into account during the due care provided. Additionally, the organization and health beliefs of these communities are linked to ancestral medicine and to a limited openness to the biomedical system. It is for this reason that it is necessary to create communication bridges that allow for fair access to quality health care for these indigenous communities [13,30].

In a bibliographic review conducted by Brewster and Morris (2015) on indigenous child health, they signal some infectious diseases suffered by indigenous babies and children, namely: diarrheas, parasitosis, respiratory diseases including otitis media, and urinary tract infections. Additionally, these diseases come along with problems such as low birth weight, malnutrition, and/or iron deficiency [31]. Fever is one of the common signs which appear in diseases, especially in infectious ones.

According to our study, the fever diagnosis by the healers is by direct touch on the skin; thermometers are seldom used, one of the reasons being that they are not able to use them. In the NICE (National Institute for Health and Care Excellence) guides on the initial management of children with fever published in 2019, and in the Primary Care Pediatrics algorithms of the Spanish Association of Primary Care Pediatrics (Asociación Española de Pediatría de Atención Primaria, AEPap) 2019, the use of a thermometer is mentioned for the detection of fever [32,33,34].

The identification of signals such as alterations in the color of the skin, irritability, inactivity, sleepiness, signs of dehydration, increases in respiratory rate, and/or intercostal retraction are the main elements signaled as alarm signs by the guides for the management of fever such as the ones by the NICE and by the AEPap. Similarly to the biomedical guides, traditional healers focus on signs of dehydration, inactivity, irritability, and sleepiness, but they lack a flow or flow diagram to follow. Additionally, their main tool is patient follow-up at their homes, which allows greater surveillance and, in case new signs and symptoms appear (or if those already present are exacerbated), knowing how to act promptly [33,34].

The use of flowcharts, algorithms or management guides in medicine ease the decision-making process according to the clinical condition, whereas in the healers, knowledge is inherited (from the father to the son), or they acquire it working with the community healer [29]. However, they lack tools for decision-making such as those of biomedicine.

One of the results of this research was the creation of a culturally adapted flowchart for the healers to identify alarm signs in children with fever and, if necessary, to transfer them to the health system. Openness to learning and teamwork could be observed in the healers. Additionally, the culturally adapted flowchart was used by 50% of the healers in a period of one year, which shows the importance of training this population. The interactions of the healers with the medical professionals and means of training sessions will allow the creation of an atmosphere of respect, equality, and trust. All these parameters will improve teamwork and the health conditions of the indigenous communities [6,30,35].

One of the limitations of this study lies in the fact that it only collects information from focus groups, when deeper data collection would be expected with interviews. In addition, it does not establish any difference between the cosmovision of the Shuar and Kichwa nationalities. On the other hand, the following aspects can be mentioned among its strengths: the ongoing relationship between the research team and the healers, apart from follow-ups by means of assessments of the training and the implementation of the flowchart for the management of children with fever.

Finally, in the beginning, 60 healers who worked on the investigation were trained, but upon request from the Public Health Ministry of Ecuador, the rest of the indigenous communities of southern Ecuador were trained on the instrument’s use. The total of healers trained until now is 120.

## 5. Conclusions

This study has identified the needs of healers in the management of children with fever, highlighting the importance for them to recognize the alarm signs that can complicate the patient’s condition, as well as that they recognize the need and situations where they must transfer the child to the biomedical health system.

The results show that there are obstacles interfering with an adequate articulation between the traditional and the biomedical health systems and suggest the development of training programs based on the needs of traditional healers. They also suggest the implementation of culturally adapted flowcharts into the local health guides, as well as addressing the most pressing problems of these populations.

## Figures and Tables

**Figure 1 ijerph-20-03923-f001:**
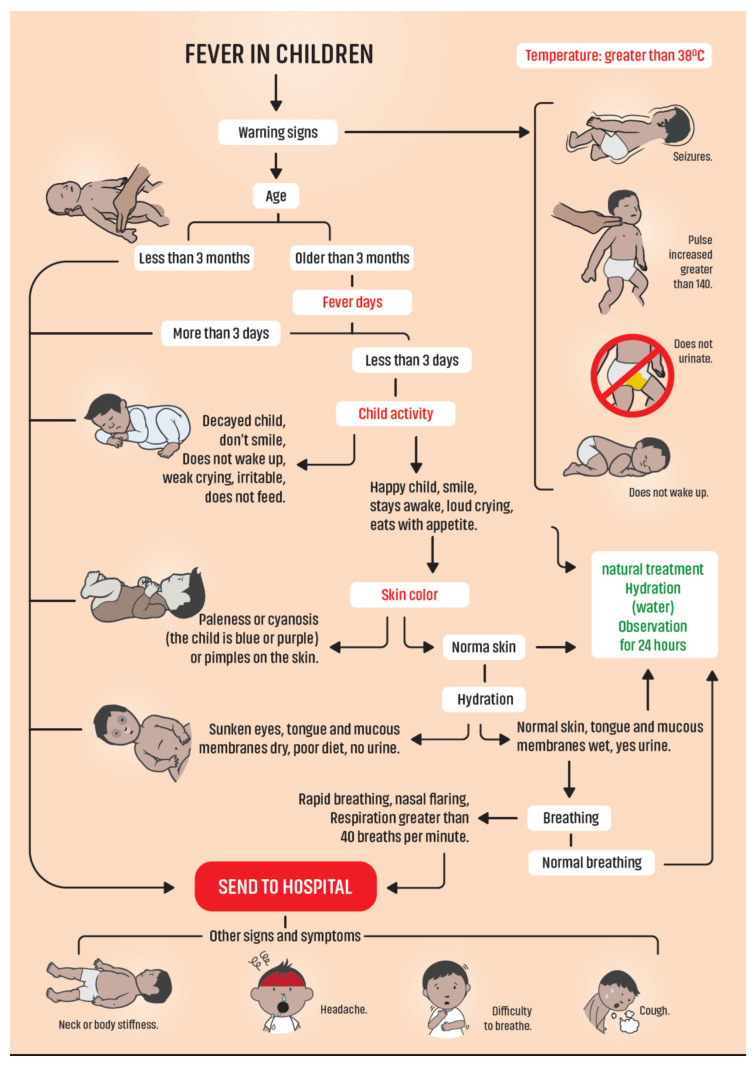
Flowchart of children with fever.

**Table 1 ijerph-20-03923-t001:** Focus group, clinical cases of “children with fever”.

Case 1:
A mother brings her 5-year-old child to the doctor, she noticed that the girl started to cough yesterday; she seems to have normal energy and plays well with her brothers, but she feels hot and eats less than normally.
Which questions will you ask the mother?
Which observations will you make of the child?
Which exams will you perform?
What do you think is the most likely cause of the problem?
Which actions (treatment) will you give?
What will you tell the mother?
How will you do the follow-up?
**Case 2:**
You are asked to give your advice on the case of a 4-year-old boy who has been ill for 5 days now. He is coughing a lot and feels very hot. The mother already gave him a medication for fever but the fever came back after a few hours. The child is very tired and wants to sleep all the time.
Which questions will you ask the mother?
Which observations will you make in the child?
Which exams will you perform?
What do you think is the most likely cause of the problem?
Which actions (treatment) will you give?
What will you tell to the mother?
How will you do the follow-up?
What is different compared to the first case?
**Case 3:**
The mother of a 7-year-old boy calls you to her house, her son suddenly started to vomit an hour ago. He is very irritable and seems a bit confused when you talk to him. He feels hot and is breathing faster than normally.
Which questions will you ask the mother?
Which observations will you make in the child?
Which exams will you perform?
What do you think is the most likely cause of the problem?
Which actions (treatment) will you give?
What will you tell to the mother?
How will you do the follow-up?
What is different compared to the first case?

**Table 2 ijerph-20-03923-t002:** Evaluation guide.

Have you attended children with fever in the last year?
*Yes/No*
**How many children with fever have you attended in the last year?**
(number)
**Have you had complications in the care of a child with a fever and what have they been?**
*Yes/No*
**If your answer is yes, what were these?**
**Have you had to transfer any patient to the hospital?**
*Yes/No*
**How the circumstances were for the transfer to the hospital:**
*Transport*
*Acceptance at the hospital*
*Cooperation and collaboration of the hospital staff*
*Communication hospital staff for the midwife or healer*
**Do you think that the flowchart of children with fever is easy to understand and handle? Rate from 1 to 5. Being 1 nothing understandable and nothing useful and 5 very useful**
*1. I don’t understand and not useful*
*2. It is understood little and little useful*
*3. It is understood moderately and is moderately useful*
*4. It is understood and useful*
*5. Very useful*

**Table 3 ijerph-20-03923-t003:** Socio-demographic characteristics and focus groups.

Characteristic	Healers (N = 65)
	N (%)
**Gender**	46 (70.8) 19 (29.2)
Female	
Male	
**Age (years)**	
<30	9 (13.8)
30–49	33 (50.8)
50–69	21 (32.3)
>70	2 (3.1)
**Ethnicity**	
Kichwa	54 (83.1)
Mestizo	6 (9.2)
Shuar	5 (7.7)
**Education**	
Illiterate	4 (6.2)
Primary	15 (23.1)
High school	40 (61.5)
Higher	6 (9.2)
**Healer**	
Yachay/Uwishin	9 (13.8)
Midwife/parteras	35 (53.8)
Herb-healer (“Hierbateros”),	14 (21.5)
Bonesetter (“sobadores”).	7 (10.7)

**Table 4 ijerph-20-03923-t004:** Categoy and Subcategories.

Category	Subcategories
Caring for children with fever in indigenous communities	Steps in caring for a child with fever.
Alarm signs in children with fever	Signs and symptoms Physical examination of the patient
Patient follow-up	Healing rituals or treatments
Referral to the Health System	Obstacles to referral to the patients

## Data Availability

The datasets generated and/or analyzed during the current study are not publicly available because they contain the sensitive personal information of participants. The informed consent grants the confidentiality of the participants’ data. However, the datasets are available from the corresponding author upon reasonable request.

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
