# Peer review of "Improving the Management of Children with Fevers by Healers in Native Rural Areas in the South of Ecuador"

_ijerph, 2023, doi:10.3390/ijerph20053923_

Round 1
Reviewer 1 Report
This is a very interesting and valuable paper. The need for culturally appropriate primary healthcare that respects traditional practices is great. Tools to build bridges between traditional carers and help enhance their work are vital. Congratulations to the authors for this project. I have placed some comments in the attached file.
Also for your consideration but not a requirement I want to let you know that in my home country (Australia) the word Indigenous when used in context of people is always spelled with a capital I as a respect. I am not sure if this is common practice in your country but thought you might like to consider.

Author Response
January, 2023
Nora Shang
International Journal of Envioronmental Research and Public Health
Thanks, by consider our manuscript entitled “Improving management of children with fever by healers in native´s rural areas of Ecuador”
In our manuscript several changes have been made. The reviewers´ suggestions were made as some additional changes.
Reviewer 1
- There is a Word or two missing from the sentence. It is not complete
We have decided to add the missing word
In order to ease discussions at various levels, different types of healers participated in the research
- How did these focus groups meet the principles for PAR. Were any participants involving the questions or contributing to he clinical case studies? were any participants involved in conducting and moderating groups? canvassing participants opinions is no enough to reflect genuine PAR research practice
The healers participated in the elaboration of the clinical cases and the guide of focus groups. Suggestions were added to the document.
In the focus groups clinical cases on paper were presented to and discussed by the participants, to help understand the management of children with fever from the perspective and context of traditional healers. The clinical cases were written clearly and concisely, the simplicity of the language allowed the participants to have better interaction and understanding. The main idea ​​focused on the warning signs in children with fever. Each focus group was conducted with the help of one moderator (researcher) and one observer (researcher). The guide for the focus groups was elaborated by the team of researchers and the healers. This consisted of a clinical case focusing on the management of children with fever in rural indigenous communities (Table 1). In addition, their informed consent and authorization to audio-record the discussions were asked for.
- There is a word missing at the end of this sentence
We add the missing word
Reflecting on the action in each of the clinical cases, and interpreting through a clear and interpretative discourse
- the questions you asked are not well designed to explore feelings, attitudes or motivations. They are very procedural questions
we eliminated feeling, attitudes or motivation by procedural
In the results, the perception of the healers was presented because it seeks to know the ideas, , skills and aptitudes of the healers about management of children with fever, and with this content, design a culturally adapted tool (flow chart) that allows them to strengthen their work and create communication bridges with the biomedical system to improve health in indigenous communities.
- Did the healers have an opportunity to co-desing or comment on the creations of the flowchart?
The healers who give us the feedback and suggest some changes
The researchers elaborated the designs and then presented them to the healers in several sessions where they proposed various changes such as warning signs and words such as purple or bluish skin were increased to facilitate the interpretation of cyanosis
- how were participants involved in creating the flowcharts. How was local design incorporated. was the graphic designer familar with the cultural group and accepted by them
We add the about disign and role of healers
The design of the graphics was based on the Saraguro indigenous culture, which is the participants' place of birth and residence. The designer carried out a prior revision of the indigenous group
- reviewer It is clear why the word beliefs is in quote marks in this sentence
We delete quote marks
8). Is this the correct word in this context or shuld it be quality?
The correct Word is quality

Reviewer 2 Report
Dear authors
The study is interesting; however, I highlight several points that should be considered.
1. The organisation of the methodology, results and discussion should be orderly and methodical. I want to suggest that the authors follow the SRQR or CORQ guidelines (https://www.equator-network.org/) to be able to present all the details of a work of this nature.
2. On the other hand, once you read the whole document, it seems that it is a mixed study, i.e. quantitative (they present percentages of adherence to the strategy) and qualitative.
3. Since it is a qualitative study, several essential points are missing, such as a description of the theoretical framework, a description of the criteria for discourse saturation, quality control criteria for qualitative research, the definition of theoretical categories, a definition of the population and type of sampling (the authors point out the type of sampling for healers but not for health professionals).
4. The authors should justify the focus group as the qualitative research technique that was used. It seems instead that the method used was the "discussion group".
5. Line 110-111 Incomplete statement. Please complete the idea. On the other hand, the authors should explain the reference in that sentence.
6. References: references number 23 and 24 should be actualized.
Author Response
January, 2023
Nora Shang
International Journal of Envioronmental Research and Public Health
Thanks, by consider our manuscript entitled “Improving management of children with fever by healers in native´s rural areas of Ecuador”
In our manuscript several changes have been made. The reviewers´ suggestions were made as some additional changes.
Reviewer 1
- There is a Word or two missing from the sentence. It is not complete
We have decided to add the missing word
In order to ease discussions at various levels, different types of healers participated in the research
- How did these focus groups meet the principles for PAR. Were any participants involving the questions or contributing to he clinical case studies? were any participants involved in conducting and moderating groups? canvassing participants opinions is no enough to reflect genuine PAR research practice
The healers participated in the elaboration of the clinical cases and the guide of focus groups. Suggestions were added to the document.
In the focus groups clinical cases on paper were presented to and discussed by the participants, to help understand the management of children with fever from the perspective and context of traditional healers. The clinical cases were written clearly and concisely, the simplicity of the language allowed the participants to have better interaction and understanding. The main idea ​​focused on the warning signs in children with fever. Each focus group was conducted with the help of one moderator (researcher) and one observer (researcher). The guide for the focus groups was elaborated by the team of researchers and the healers. This consisted of a clinical case focusing on the management of children with fever in rural indigenous communities (Table 1). In addition, their informed consent and authorization to audio-record the discussions were asked for.
- There is a word missing at the end of this sentence
We add the missing word
Reflecting on the action in each of the clinical cases, and interpreting through a clear and interpretative discourse
- the questions you asked are not well designed to explore feelings, attitudes or motivations. They are very procedural questions
we eliminated feeling, attitudes or motivation by procedural
In the results, the perception of the healers was presented because it seeks to know the ideas, , skills and aptitudes of the healers about management of children with fever, and with this content, design a culturally adapted tool (flow chart) that allows them to strengthen their work and create communication bridges with the biomedical system to improve health in indigenous communities.
- Did the healers have an opportunity to co-desing or comment on the creations of the flowchart?
The healers who give us the feedback and suggest some changes
The researchers elaborated the designs and then presented them to the healers in several sessions where they proposed various changes such as warning signs and words such as purple or bluish skin were increased to facilitate the interpretation of cyanosis
- how were participants involved in creating the flowcharts. How was local design incorporated. was the graphic designer familar with the cultural group and accepted by them
We add the about disign and role of healers
The design of the graphics was based on the Saraguro indigenous culture, which is the participants' place of birth and residence. The designer carried out a prior revision of the indigenous group
- reviewer It is clear why the word beliefs is in quote marks in this sentence
We delete quote marks
8). Is this the correct word in this context or shuld it be quality?
The correct Word is quality
Second reviewer
The study is interesting; however, I highlight several points that should be considered.
- The organisation of the methodology, results and discussion should be orderly and methodical. I want to suggest that the authors follow the SRQR or CORQ guidelines (https://www.equator-network.org/) to be able to present all the details of a work of this nature.
We review the COREG guide
This paper follows the consolidated criteria for reporting qualitative research (COREQ) guidelines for reporting qualitative research
- On the other hand, once you read the whole document, it seems that it is a mixed study, i.e. quantitative (they present percentages of adherence to the strategy) and qualitative.
Thanks for your suggestions, but It is a Participatory active reseach. We presented percenages of adherence to the strategy, because PAR has four parts, one of them is the evaluation of the implementation, y we prefer present porcentajes because it give us precise information on the use of the tool and the reference system
- Since it is a qualitative study, several essential points are missing, such as a description of the theoretical framework, a description of the criteria for discourse saturation, quality control criteria for qualitative research, the definition of theoretical categories, a definition of the population and type of sampling (the authors point out the type of sampling for healers but not for health professionals).
We add about saturation, quality control, definition theoretical categories and definition of the population
The saturation criteria of the discourse were the repetition of the topics and that no new information was shared. Despite the criteria, saturation is an art that will depend on the type of questions and how they are asked to obtain different answers. In addition, the nuance of an answer can have different interpretations; here, the importance of triangulation arises.
Quality control criteria for qualitative research
Researchers have reviewed the COREQ guide for research qualityThe definition of theoretical categories
| Category | Subcategories |
| Caring for children with fever in indigenous communities | Steps in caring for a child with fever. |
| Alarm signs in children with fever | Signs and symptomsPhysical examination of the patient |
| Patient follow-up | Healing rituals or treatments |
| Referral to the Health System | Obstacles to referral to the patients |
A definition of the population and type of sampling
We chose to recruit indigenous groups from Southern Ecuador because: a) these communities are the center of meeting for indigenous celebrations, and b) these communities belong to native Ecuadorian people
Purposive sampling was used to recruit the participants.
- The authors should justify the focus group as the qualitative research technique that was used. It seems instead that the method used was the "discussion group".
We add about discussion groupfocus groups, the main idea of ​​this methodology and technique is to generate a dialogue on a specific topic and also observe and pay attention to the attitudes of the study participants.
- Line 110-111 Incomplete statement. Please complete the idea.
We completed statement
In order to ease discussions at various levels, different types of healers participated in the research
- References: references number 23 and 24 should be actualized.
We changes the references.
- Otani T. [What Is Qualitative Research?]. Yakugaku Zasshi. 2017;137(6):653-658. Japanese. doi: 10.1248/yakushi.16-00224-1. PMID: 28566568.
- Moser A, Korstjens I. Series: Practical guidance to qualitative research. Part 3: Sampling, data collection and analysis. Eur J Gen Pract. 2018 Dec;24(1):9-18. doi: 10.1080/13814788.2017.1375091. Epub 2017 Dec 4. PMID: 29199486; PMCID: PMC5774281.
Third reviewer
This study is important and covers a topic that does not receive a lot of attention, but is common in resource-limited settings. Overall there are some issues with language, typos, and incomplete sentences throughout that need to be more closely reviewed.
- Intro has some sentences and background that seem incomplete, for example: The first sentence references the MDGs, but does not say by when the goal of mortality reduction is to occur.
We add the period
One of the Millennium Development Goals of the United Nations (UN) was to reduce mortality in children below 5 years old by two thirds between 1990 and 2015. The UN 2015 report signals that the mortality rate has reduced by half, from 90 to 43 deaths per 1,000 live-born children between 1990 and 2015.
- The second paragraph and others could be more concise, e.g.: "Child mortality was 5.1% higher in indigenous children (reference)." There are several other examples of this. The lit review is also not comprehensive and while it is noted that there are examples of healers being detrimental to the management of biomedical illnesses, there are no examples of how healers have improved illness treatment or complemented the biomedical system, of which several examples exist.
We add the paragraph
A clear example is the study carried out in the Kilosa and Handen districts of Tanzania which demonstrated the critical role of traditional healers in providing "biomedically accepted first aid," leading to lowering of body temperature and thus increased chances of survival of patients with Malaria (children) [9]
Makundi EA, Malebo HM, Mhame P, Kitua AY, Warsame M. Role of traditional healers in the management of severe malaria among children below five years of age: the case of Kilosa and Handeni Districts, Tanzania. Malar J. 2006 Jul 18;5:58. doi: 10.1186/1475-2875-5-58. PMID: 16848889; PMCID: PMC1540433.
Methods:
- It would help greatly to describe the context of the study. Where did the study occur? What is the medical landscape here (i.e. it is a rural area located [location details]. The health system is pluralistic, with patients using government-funded, traditional, and privately-funded services etc.). Some of this may also be more appropriate in the background if not specific to the study area/population itself.
We add the suggestions
The traditional healers selected were midwives, chamanes ("Yachay"/"Uwishin" or "visionarios") and herb healers ("Hierbateros"), from the Saraguro and Cuenca region. We chose to recruit indigenous groups from Southern Ecuador because: a) these communities are the center of meeting for indigenous celebrations, and b) these communities belong to native Ecuadorian people. All the participants were renowned healers in the communities and expressed experience and interest in the study
- It is also unclear how the study population was recruited. Though it was purposive, how did the researchers find participants? Is there a registry? Was it advertised? Word of mouth?
We add the suggestions
In the first phase of the research the approach to the indigenous community was carried out specifically to the traditional healers through two primary health care technicians from the same community. We chose to recruit indigenous groups from Southern Ecuador because: a) these communities are the center of meeting for indigenous celebrations, and b) these communities belong to native Ecuadorian people.
- Focus group methods also need to be clarified. Who was the moderator? Are they a member of the community? Fluent in the language? Did all participants speak the same primary language/dialect?
We add the suggestions
Focus group
Each focus group was conducted with the help of one moderator (researcher) and one observer (researcher).
Language
Additionally, participants needed to be able to communicate in Spanish, although for the healers it was not their mother language which explains the grammatical mistakes that you might find in the quotes.
- Additional information on the preparation of content is also required. Was the recorded content transcribed and then analyzed? Was it translated prior to analysis? Who were the researchers who did the coding (exact names not needed, but would help to know if any were local/able to review for cultural context etc.)?
We add the suggestions
Two of the researchers speak Spanish and come from the southern region of Ecuador, so they know the worldview of the place.
For Phase 2, I am a bit confused about flow chart development. Based on responses by healers, the researchers then made a flow chart? also who are the medical professionals and how were they infolved in this process? Who was present for these group sessions? I am unclear as to how the qualitative results were translated into a flow chart. What makes this flowchart "culturally appropriate?"
We add the suggestions
In this second phase, we sought to create knowledge based on the observation results in order to solve the known problems. The qualitative results (of phase 1) were sent and presented again to all the involved researchers for a critical analysis. In three reflexive group sessions, a flowchart of children with fever was developed. As the results took into account the opinion of healers , we can say it is a culturally adapted flowchart. Reflexibility was an important cornerstone in the validity and transparency of the research and allowed the researchers to have constant interaction with the phenomenon under study [28]. Goal was to establish a joint strategy to improve the practices and the inter-cultural communication bridges. Additionally, the strategy focused on the detection of alarm signs in children with fever and to activate a referral system in a rural health community team. The researchers created the flowcharts based on the results obtained in phase 1, and the design was done with the help of a graphic designer. The design of the graphics was based on the Saraguro indigenous culture, which is the participants' place of birth and residence. The designer carried out a prior revision of the indigenous group. Finally, a last reflexive group session was held with the traditional healers; the flowchart was presented and discussed. The healers made some suggestions on the content and the graphics of the flowchart presented such us warning signs and add words like purple or bluish skin were increased to facilitate the interpretation of cyanosis
- Again for the action plan, would help to know what types of health professionals were involved and how they were selected. For the evaluation guide, it would help to know what is considered a complication of fever and if this definition was communicated to those surveyed. Also, how was the survey administered?
We add the suggestionsThe health professionals involved in the training were the researchers who are one family doctor, one pediatrician and one obstetrician-gynecologists
One year after the training of the traditional healers and health professionals, the researchers approached the indigenous communities again to assess the implementation of the culturally-adapted flowchartin children with fever through of a survey.
Results:
- Table 3 with the FG breakdown is not necessary. Details on focus group duration and code saturation should be in methods.
We delete the details on focus group
The themes could be more specific. I think the first theme of “caring for children” more accurately reflects use of traditional diagnoses to interpret symptoms and use of traditional treatments and assessments. The “patient follow up” theme seems more to describe use of traditional treatments and the role the healer plays in continuing to be involved in the child’s care (including going to their home if necessary). The “referral theme” seems to describe primarily the obstacles healers may face and interactions with the biomedical system. Overall, I think the themes and sub-themes could be described a bit more in-depth.
We described a bit more in depth
- For the Action plan, how was content of the workshops determined. Again, how was this culturally adapted? Was there any sort of knowledge assessment or skills check to ensure healers felt comfortable with these skills or took away the intended learning points? How many healers participated in this phase?
We add the suggestions
A series of ten workshops were held on the management of children with fever flowchart facilitated by the researchers. These workshops were developed in culturally adapted skills laboratories; each group consisted of 6 people and lasted 2 hours, all healers were trained (n=56) The main topics developed in the workshops were:
- The use of a thermometer and its interpretation,
- Recognition of alarm signs and patient management (steps from the flowchart)
- Patient referral system.
The content of the workshops responds to the needs of healers observed during the investigation of children with fever.
- Evaluation: why are the rest of the results from this area not reported or any table/numbers given? What percentage of the healers who participated in the workshops were evaluated? What does it mean that the healer “used the flowchart” and how did this ultimately affect their management?
We add the suggestions
Most of the transfers (70%) were accompanied by the healer. Finally, 60% of the healers think that the flowchart is moderately understandable and useful, and 5% think that it is poorly understood and not very useful.
- For the discussion, would start with the findings of the current study and how they are innovative/pertinent, then move on to similar studies and next steps. Some of the topics discussed, such as the that there is a “limited openness to the biomedical system” are not discussed in results or highlighted as themes. It is also unclear how the healers management differs from that of others working in low resource settings or what would be standard of care in this area as some of the danger signs etc. do not seem that different from what the World Health Organization recommends for Integrated Management of Childhood Illness.
We made some changes to the discussion.
We hope that you find this new version suitable for publication. Thank you for your consideration.
Sincerely,
Md. Estefanía Bautista-Valarezo
mebautista@utpl.edu.ec
Telephone: +593 987356056
Departamento de Ciencias de la Salud
Universidad Técnica Particular de Loja (UTPL)
San Cayetano alto s/n, CP: 1101608,
Loja, Ecuador.

Reviewer 3 Report
This study is important and covers a topic that does not receive a lot of attention, but is common in resource-limited settings. Overall there are some issues with language, typos, and incomplete sentences throughout that need to be more closely reviewed.
Intro has some sentences and background that seem incomplete, for example: The first sentence references the MDGs, but does not say by when the goal of mortality reduction is to occur. The second paragraph and others could be more concise, e.g.: "Child mortality was 5.1% higher in indigenous children (reference)." There are several other examples of this. The lit review is also not comprehensive and while it is noted that there are examples of healers being detrimental to the management of biomedical illnesses, there are no examples of how healers have improved illness treatment or complemented the biomedical system, of which several examples exist.
Methods:
It would help greatly to describe the context of the study. Where did the study occur? What is the medical landscape here (i.e. it is a rural area located [location details]. The health system is pluralistic, with patients using government-funded, traditional, and privately-funded services etc.). Some of this may also be more appropriate in the background if not specific to the study area/population itself.
It is also unclear how the study population was recruited. Though it was purposive, how did the researchers find participants? Is there a registry? Was it advertised? Word of mouth?
Focus group methods also need to be clarified. Who was the moderator? Are they a member of the community? Fluent in the language? Did all participants speak the same primary language/dialect?
Additional information on the preparation of content is also required. Was the recorded content transcribed and then analyzed? Was it translated prior to analysis? Who were the researchers who did the coding (exact names not needed, but would help to know if any were local/able to review for cultural context etc.)?
For Phase 2, I am a bit confused about flow chart development. Based on responses by healers, the researchers then made a flow chart? also who are the medical professionals and how were they infolved in this process? Who was present for these group sessions? I am unclear as to how the qualitative results were translated into a flow chart. What makes this flowchart "culturally appropriate?"
Again for the action plan, would help to know what types of health professionals were involved and how they were selected.
For the evaluation guide, it would help to know what is considered a complication of fever and if this definition was communicated to those surveyed. Also, how was the survey administered?
Results:
Table 3 with the FG breakdown is not necessary. Details on focus group duration and code saturation should be in methods.
The themes could be more specific. I think the first theme of “caring for children” more accurately reflects use of traditional diagnoses to interpret symptoms and use of traditional treatments and assessments. The “patient follow up” theme seems more to describe use of traditional treatments and the role the healer plays in continuing to be involved in the child’s care (including going to their home if necessary). The “referral theme” seems to describe primarily the obstacles healers may face and interactions with the biomedical system. Overall, I think the themes and sub-themes could be described a bit more in-depth.
For the Action plan, how was content of the workshops determined. Again, how was this culturally adapted? Was there any sort of knowledge assessment or skills check to ensure healers felt comfortable with these skills or took away the intended learning points? How many healers participated in this phase?
Evaluation: why are the rest of the results from this area not reported or any table/numbers given? What percentage of the healers who participated in the workshops were evaluated? What does it mean that the healer “used the flowchart” and how did this ultimately affect their management?
For the discussion, would start with the findings of the current study and how they are innovative/pertinent, then move on to similar studies and next steps. Some of the topics discussed, such as the that there is a “limited openness to the biomedical system” are not discussed in results or highlighted as themes. It is also unclear how the healers management differs from that of others working in low resource settings or what would be standard of care in this area as some of the danger signs etc. do not seem that different from what the World Health Organization recommends for Integrated Management of Childhood Illness.
Overall I think this work is important, but the communication of all aspects of the study needs to be improved. I was left with many questions after reading it.
Author Response

(The authors gave the same response as above.)
